# Identifying Clinicopathological Factors Associated with Oncotype DX^®^ 21-Gene Recurrence Score: A Real-World Retrospective Cohort Study of Breast Cancer Patients in Quebec City, Canada

**DOI:** 10.3390/jpm11090858

**Published:** 2021-08-28

**Authors:** Simon Gagnet, Caroline Diorio, Louise Provencher, Cynthia Mbuya-Bienge, Julie Lapointe, Claudya Morin, Julie Lemieux, Hermann Nabi

**Affiliations:** 1Axe Oncologie, Centre de Recherche du CHU de Québec-Université Laval, Québec, QC G1R 3S3, Canada; simon.gagnet@gmail.com (S.G.); caroline.diorio@crchudequebec.ulaval.ca (C.D.); provencher.louise@me.com (L.P.); cynthia.mbuya-bienge.1@ulaval.ca (C.M.-B.); julie.lapointe@crchudequebec.ulaval.ca (J.L.); julie.lemieux@crchudequebec.ulaval.ca (J.L.); 2Département de Médecine Sociale et Préventive, Faculté de Médecine, Université Laval, Québec, QC G1V 0A6, Canada; 3Centre de Recherche sur le Cancer, Université Laval, Québec, QC G1V 0A6, Canada; 4Centre des Maladies du Sein Deschênes-Fabia, CHU de Québec-Université Laval, Québec, QC G1S 4L8, Canada; claudya.morin.med@ssss.gouv.qc.ca

**Keywords:** adjuvant chemotherapy, breast cancer subtype, cancer recurrence risk score, clinical decision making, gene expression profiling, lifestyle risk factors, precision medicine

## Abstract

Gene expression profiling tests such as the Oncotype DX (ODX) 21-gene recurrence score (RS) assay is increasingly used in clinical practice to predict the risk of recurrence and support treatment planning for early-stage breast cancer (BC). However, this test has some disadvantages such as a high cost and a long turnaround time to get results, which may lead to disparities in access. We aim to identify clinicopathological factors associated with ODX RS in women with early-stage BC. We conducted a retrospective cohort study of women identified in the medical database of the Deschênes-Fabia Breast Disease Center of Quebec City University, Canada. Our sample consists of 425 women diagnosed with early-stage BC who have obtained an ODX RS between January 2011 and April 2015. The ODX RS has been categorized into three levels as originally defined: low (0–17), intermediate (18–30), and high (>30). The mean RS was 17.8 (SD = 9.2). Univariate analyses and multinomial logistic regressions were performed to identify factors associated with intermediate and high RS compared with low RS. A total of 237 (55.8%) patients had low RS, 148 (34.8%) had intermediate RS, and 40 (9.4%) had high RS. Women with progesterone receptor (PR)-negative (ORs ranging from 3.51 to 10.34) and histologic grade II (ORs ranging from 3.16 to 23.04) tumors were consistently more likely to have intermediate or high RS than low RS. Similar patterns of associations were observed when the RS was categorised using redefined thresholds from (i.e., from the TAILORx study or dichotomized). This study provides evidence suggesting that histologic grade and PR status are predictive factors for intermediate or high RS in women with early-stage BC. If these results are confirmed in future studies, considering these clinicopathological factors could spare women the need to get such a test before the beginning of a possible adjuvant therapy. This option could be considered in settings where the cost of testing is an issue.

## 1. Introduction

Gene expression profiling tests are increasingly used in clinical practice to predict the risk of recurrence and personalize treatment for early-stage breast cancer (BC) [1]. One of such tests is the Oncotype DX® (ODX), which is a commercially available assay developed by Genomic Health Inc. (Redwood City, California, USA) and designed to measure the 10-year risk of tumor recurrence in early-stage BC at the time of initial diagnosis. The test evaluates the expression of 21 targeted genes (16 cancer-related and 5 reference genes) and provides a single recurrence score (RS), which ranges between 0 and 100 [2,3]. Initially developed among women with estrogen receptor-positive (ER+), epidermal growth factor receptor 2-negative, and lymph node-negative breast tumors, the RS is used to stratify patients into one of the three original risk categories: low (RS < 18), intermediate (RS 18–30), or high (RS > 30), and to help identify which women are most likely to benefit from adjuvant chemotherapy [4,5].

The use of RS to guide treatment decision making has been integrated into several clinical guidelines including those from the American Society of Clinical Oncology (ASCO) [4], the National Comprehensive Cancer Network (NCCN) [5], the European Society of Medical Oncology (ESMO) [6], and the St Gallen International Expert Consensus guidelines [7]. In Canada, the use of ODX test is funded through provincial programs and covers women who meet eligibility criteria [8]. In the province of Quebec in particular, ODX test was recommended to be used in oncology care by several committees and reimbursed by the *Régie de l’assurance maladie du Quebec* (the government health insurance board) for any request that meets the authorization and reimbursement mechanism for a medical biology analysis not available in Quebec [2].

In theory, the test is available and accessible to any patient who meets the eligibility criteria. However, this is far from being the case, and there is compelling evidence suggesting that the utilization of ODX varies greatly between oncologists, populations, and locations [8,9,10,11]. Several factors have been suggested to explain these disparities in the utilization and the availability of ODX. One of these factors is the cost of the test itself which amounts to more than CAD 4000 per patient [12]. Studies conducted in the United States (US) and in Europe estimated that ODX is used for only 20 to 30% of eligible patients due to reduced access and reimbursement policies [13,14,15]. Another factor is the turn-around time to get results, which could be long given that the test is exclusively performed in the US by Genomic Health Inc.

As a consequence, several studies sought to know whether the ODX RS could be predicted by BC clinicopathological factors [16,17,18,19]. Histologic grade and progesterone receptor (PR) levels were the most consistent factors associated with high RS [16,17,18,19,20], even though associations were also reported for ER level [16] and tumor size [19]. It should be noted that some of these studies had several limitations including the restriction to only stage 1 BC patients [16], the lack of consideration of hormone receptors levels [17], and the combination of patients with intermediate and low RS in comparison with those with high RS [16,17,18,19]. 

In addition, to our knowledge, no study evaluated the associations between the RS and lifestyle risk factors for BC such as tobacco use, alcohol consumption, and body mass index (BMI) alongside classical clinicopathological factors. There is evidence showing that smoking may induce DNA methylation and alter the expression profile of certain genes associated with BC risk [21,22].

Thus, the main objective of this study was to identify clinicopathological and potential lifestyle factors associated with the 21-gene ODX RS in women with early-stage BC. We also aim to examine the adequacy between RS levels and adjuvant chemotherapy decision making.

## 2. Methods

### 2.1. Study Design and Population

We conducted a retrospective cohort study that included all women included in the Deschênes-Fabia Breast Disease Center (CMS-DF) medical database who were diagnosed with early-stage (i.e., stage I and II) invasive BC and who obtained an ODX RS between January 2011 and April 2015. The CMS-DF serves the great region of Quebec City and the eastern part of the Quebec province. The center treats approximately 1000 new BC cases each year, which represents about 15% of all BC cases in the province and records more than 30,000 visits annually, making it one of the largest BC centers in Canada.

### 2.2. Variables

#### 2.2.1. Oncotype DX Recurrence Score 

We categorized the ODX RS into three risk categories as done previously [23]. Patients with a RS of 0–17 were considered at low risk of recurrence. Those with a RS of 18–30 were considered to have an intermediate risk. Finally, those with a RS of >30 were considered at high risk.

#### 2.2.2. Clinicopathological and Lifestyle Risk Factors

Clinicopathological and lifestyle risk factors were obtained from the CMS-DF database. Clinicopathological factors included hormone receptor levels (i.e., estrogen receptors (ER), progesterone receptors (PR)), human epidermal growth factor receptor (HER2) status, tumor grade, histologic type, size and stage of the tumor, lymph node status, menopausal status, parity, age at first menstruation, age at diagnosis, and BMI. Data on lifestyle habits such has smoking status and alcohol consumption were collected from a questionnaire filled by women during their first consultation at the clinic. More details regarding the CMS-DF database constitution are provided in a previous publication [24]. ER and PR levels were obtained with a qualitative evaluation of the immunohistochemically (IHC) receptors. The IHC test uses specific antibodies monoclonal receptors to detect the presence of hormone receptors and determine whether a tumor is positive or negative for these hormonal receptors. IHC test results are usually reported in percentage of nuclei linked with the antibody and are classified according to the following categories: <1, 1–90 and 90–100. HER2 status was either obtained by IHC testing, which evaluates the amount of HER2 protein in a cancerous cell or by in situ hybridization with fluorescence (FISH), which evaluates the number of HER2 gene copies in a cancerous cell. HER2 status was categorized into positive, negative, or equivocal. Continuous variables were, respectively, categorized as follows: age at diagnosis (≤50 and >50 years), age at first menstruation (<12, 12–14 and ≥15 years), tumor size (≤20 and >20 mm), and BMI (<25, 25–29.9 and ≥30 kg/m^2^). Smoking status was coded into two categories (smoker and non-smoker) and alcohol consumption was coded into four categories (no consumption, 0.1–9.9, ≥10 and missing category) based on the number of alcohol portions consumed per week. Menopausal status was coded into two categories (yes or no). To avoid losing statistical power, tumor stage (i.e., I or II), tumor size (i.e., ≤20 mm or >20 mm), grade (i.e., I or II), and histological type (i.e., invasive ductal carcinoma or others) were dichotomized.

### 2.3. Statistical Analyses

All statistical analyses were performed using SAS software, Version 9.4 (Copyright © 2016 by SAS Institute Inc., Cary, NC, USA). Univariate associations between RS and clinicopathological and lifestyle risk factors were first evaluated using a chi-squared test. We used multinomial logistic regressions to examine associations between RS and selected clinicopathological and lifestyle risk factors. Odds ratios (ORs) and their 95% confidence intervals (CI) were reported. The explanatory variables known in the literature to be risk or prognostic factors for BC were included in the model from the outset. The other variables were considered when the chi-squared test was significant with a *p* value < 20%. In our study, missing data were not missing at random (MNAR). For alcohol consumption, we created a dummy variable modality for the missing values because the number was important (N = 37). All tests were two-sided with a 0.05 level of significance.

### 2.4. Sensitivity Analyses

To test the robustness of our results, we conducted several sensitivity analyses. First, we performed the analyses by categorizing RS according to the classification used in the TAILORx (Trial Assigning Individualized Options for Treatment (Rx)) study [25] (i.e., low RS =≤ 10, intermediate RS = 11–25, and high RS ≥ 26). These thresholds were initially defined to minimize potential under-treatment and were also used in recent studies [18,26,27]. Second, we dichotomized the RS by grouping together the low and intermediate categories as done in a previous study [19]. Third, we excluded from the sample HER2-positive BC patients and patients with a positive lymph node. These exclusions were chosen to follow clinical practice guidelines which did not initially recommend the use of ODX test for patients with HER2-positive BC or those who have a positive lymph node. Fourth, to take into account the missing values, we created a dummy variable for each variable with missing values and re-ran the analyses.

## 3. Results

The characteristics of our sample are presented in Table 1. The total sample was composed of 425 women with 75.3% aged over 50 years old. All women in our sample (100%) were ER+, 92.7% were PR+, 98.3% were HER2- and 91.3% were lymph node-negative. The RS varied between 0 and 64 with a mean RS of 17.8 (SD = 9.2). A total of 237 women (55.8%) had low RS, 148 (34.8%) had intermediate RS, and 40 (9.4%) had high RS. For each category, the mean RS were, respectively, 11.6 (SD = 4.0), 22.1 (SD = 3.5), and 37.9 (SD = 7.5). In our sample, 74.6% of women had invasive ductal carcinoma, 81.7% had a histological grade I, and 70.5% were postmenopausal at their first consultation. About half of the women (52.4%) were smokers, and 66.8% reported consuming alcohol with 58.8% drinking between 0.1 and 9.9 servings of alcohol/week and 8% ≥10 servings of alcohol/week.

Table 2 presents the distribution of our sample’s characteristics according to the RS categories. Univariate analyses revealed a significant or closed to significance relationships between RS and four clinicopathological factors: ER (*p* = 0.0661), PR (*p* < 0.0001), histologic grade (*p* < 0.0001) and histologic type (*p* = 0.0307). In addition to these factors, age (*p* = 0.1537) and menopausal status (*p* = 0.1148) were also selected as independent variables to be included in the multivariable regression models. The size of the tumor, alcohol consumption, smoking status, and BMI were not significantly associated with RS according to our threshold set at 20% level of significance.

Results of the multinomial multivariable logistic regressions are reported in Table 3. Women with PR-negative (<1 vs. 1–90: OR = 3.51, 95% CI = 1.21–10.21) and those with histologic grade II (II vs. I: OR = 3.16, 95% CI = 1.65–6.05) tumors were more likely to have intermediate RS. Similarly, women with PR-negative (<1 vs. 1–90: OR = 10.34, 95%CI = 2.60–41.15) and those with histologic grade II tumors (II vs. I: OR = 23.04; 95%CI = 8.91–59.55) were much more likely to have high RS.

Finally, we explored the effect of the RS on decision making for adjuvant chemotherapy (Table 4). We found a significant association between the RS and adjuvant chemotherapy treatment (*p* < 0.001). Thus, 98.7% of women with low RS did not receive chemotherapy. In contrast, 92.5% of women with high RS risk received chemotherapy. Among women classified as intermediate risk, 57.4% did not received chemotherapy.

Results of our sensitivity analyses revealed similar patterns of associations to the main analyses. Categorization of RS according to the thresholds used in the TAILORx Trial [25] showed that 81 (19.1%) women had low RS, 274 (64.5%) had intermediate RS, and 70 (15,4%) had high RS. Results from our multinomial logistic regressions were quite consistent with women with PR-negative (<1 vs. 1–90: OR = 1.56; 95% CI = 0.33–7.47), and those with histologic grade II (II vs. I: OR = 2.41, 95% CI = 0.90–6.49) tumors were more likely to have intermediate RS, even though these associations were not statistically significant. Like the main analyses, women with PR-negative (<1 vs. 1–90: OR = 5.77, 95% CI = 1.05–31.73) and those with histologic grade II (II vs. I: OR = 18.77, 95% CI = 6.22–56.61) tumors were more likely to have high RS. When RS was considered as a dichotomous variable, 385 (90.6%) had low/intermediate RS, and 40 (9.4%) women had high RS. In multivariable analyses, women with PR-negative (<1 vs. 1–90: OR = 3.67, 95% CI = 1.27–10.57) and histologic grade II (II vs. I: OR = 11.66, 95% CI = 5.25–25.91) tumors were more likely to have high RS. Finally, the sensitivity analysis in which patients with HER2- and lymph node-positive tumors were excluded was based on 376 women, and the overall results are comparable with those of our main analyses.

## 4. Discussion

In this study, we sought to examine the associations between BC classical clinicopathological, some lifestyle risk factors, and RS categories as measured by the ODX test in women with early-stage BC. In univariate analyses, we found four clinicopathological factors (i.e., ER and PR levels, histological grade, and type) to be associated with RS categories. However, none of the lifestyle factors considered here were associated with the RS. In multivariate analyses, we found that women with PR-negative and histologic grade II tumors were consistently more likely to have intermediate or high RS than a low RS. Finally, there was a significant association between RS and adjuvant chemotherapy treatment, coherent with current clinical recommendations.

Our results are consistent with other studies reporting associations between histologic grade, PR, and RS. An Israeli study conducted among 300 women showed that a high histologic grade and a low PR were associated with high RS [28]. In 2011, an American study showed that histologic grade and PR status in particular were also associated with high RS [29]. Recent American [20,30] and Asian studies [17,18] also provided findings in line with ours.

Several studies compared clinicopathological characteristics with TAILORx-categorized RS by combining intermediate and low categories [18,30]. Results of our sensitivity analysis using TAILORx RS categorization were in agreement with Huang and al. [18] and Orucevic and al. [30] regarding the strong associations of RS with histologic grade and PR. However, both these studies demonstrated that RS was also associated with age and tumor size. Differences in sample sizes are likely to explain these observations. Finally, the results of our sensitivity analysis, in which low and intermediate RS categories were grouped together, showed some differences with the study by Yu-Qing and al [19]. They observed a significant association between RS and the size of the tumor, which was not the case in our study.

We were surprised to find that alcohol consumption, smoking status, BMI, and age were not significantly associated with RS in our univariate analyses. However, it is important to stress the difficulty of collecting accurate data on alcohol consumption due to a social desirability bias, which might underestimate the true association. Furthermore, the choice to create a "missing data" modality allowed us not to lose any information, but can also lead to an underestimation of the association in comparison with situations where all the data are available. If we compare our sample with the smoking status of the Canadian population, we observed that 47.6% of women in our study declared that they have never smoked compared with only 30% of women in the Quebec population in 2013–2014 [31]. In regard to the BMI, our results are in agreement with various studies that did not show an association between RS and BMI [32,33]. However, one study has found RS to be higher in overweight patients with node-negative BC [34]. In addition, in our study, the variable "age" adjusted for other covariates was not significantly associated with RS, which seems to be consistent with several studies [17,19,20], even though others have reported an association [18,35]. These discrepancies could be explained by the difference in sample size or the classification of the RS. Thus, in light of contradictory results, we believe these associations deserve to be explored in future studies. 

Although not statistically significant, some of our results are still difficult to explain. The ORs for age are in different direction when intermediate RS was compared with low RS (OR = 0.77) and when high RS was compared with low RS (OR = 3.31). Thus, a woman over 50 is less likely to have intermediate RS than low RS, but she is more likely to have high RS than low RS. For the menopausal status, the association was not significant with RS, as showed by several studies [33,36], with the exception of the study by Picardo and al [34]. Similar to age, the interpretation of the association between menopausal status and RS remains challenging. Indeed, a postmenopausal woman had a higher probability of having intermediate RS rather than low RS (OR = 1.35) but had a lower probability of having high RS than low risk (OR = 0.28). On the other hand, we have highlighted an expected relationship between RS and adjuvant chemotherapy treatment, suggesting that decision making of treatment allocation is consistent with current recommendations, which is also observed in the literature [19]. However, there appears to be a decision making difficulty with adjuvant chemotherapy for women classified as intermediate risk, which the TAILORx [25] trial intended to clarify. 

Our study has some limitations. First, we used a retrospective design with a limited follow-up period. A longer follow-up would have allowed us to put our results in perspective with overall survival or disease-free survival rate, for example. The sample size may have also limited the study statistical precision, as the CIs of some of our ORs are quite large. In addition, the collection of the majority of characteristics was done at baseline while the ODX test result might be received weeks later due to the process surrounding the surgery, pathology, and laboratory timeline for analysing and returning the results. Thus, possible changes in lifestyle factors during this three- to four-month period could not be collected. Finally, several studies have explored the relationship between RS and Ki67 protein, some of which showed a significant relationship [17,20,37]. Since we did not have information on this protein in our database, its association with RS could not be explored, which appears as a potential limit in our study. Nevertheless, our results constitute an important contribution to this line of research by providing additional evidence that some clinicopathological factors, particular PR level and histologic grade, are potential predictors of RS categories. We are able to test the robustness of our findings in several sensitivity analyses and to consider for the first time relevant lifestyle BC risk factors. In addition, we were able to consider hormonal receptor values rather than status, which have been measured by HIC as recommended. Thus, our results appear plausible and in agreement with the results of studies published in the literature on this issue.

## 5. Conclusions and Perspective

Our study shows that histologic grade II and negative PR status are potential predictive factors for intermediate RS and high RS in women with invasive early-stage BC. In our study, lifestyle factors are not significantly associated with RS. Nevertheless, they deserve to be studied more in the future. The histologic grade and the PR status may be considered in advance of prescribing an ODX test. Indeed, this would allow some women, particularly those with a clear clinicopathological profile of their tumor, to bypass the ODX testing and avoid a significant wait time before the start of a possible adjuvant therapy. This finding has also the potential to reduce healthcare costs, given that performing the test might not be required for women bearing such tumor profiles. However, it is clear that further large-scale studies are needed to consolidate these results. Development and validation of accurate prediction models for ODX RS based on relevant clinicopathological factors are also needed in order to identify patients who should undergo an adjuvant chemotherapy treatment [20]. 

## Figures and Tables

**Table 1 jpm-11-00858-t001:** Characteristics of the study sample.

Characteristics	Total (N = 425)	%
Mean Recurrence Score (RS) [SD]	17.8 [9.2]	
**Recurrence score**		
Low risk (0 to 17)	237	55.8
Intermediate risk (18 to 30)	148	34.8
High risk (more than 30)	40	9.4
**Age**		
≤50	105	24.7
>50	320	75.3
**Estrogen receptor** (% cells staining positive)		
1–90	50	11.7
90–100	375	88.2
**Progesterone receptor** (% cells staining positive)		
<1	31	7.3
1–90	228	53.6
90–100	166	39.1
**Human Epidermal Growth Factor Receptor-2**		
Negative	417	98.1
Positive	2	0.5
Equivocal result	5	1.2
Missing values	1	0.2
**Histological type**		
Invasive ductal carcinoma	317	74.6
Others	108	25.4
**Tumor stage**		
I	234	55.1
II	191	44.9
**Histological grade**		
I	347	81.7
II	78	18.3
**Tumor size**		
≤20 mm	256	60.2
>20 mm	169	39.8
**Ganglion node**		
No	379	89.2
Yes	36	8.4
Missing values	10	2.4
**Smoking status**		
Non-smoker	200	47.0
Smoker	220	51.8
Missing values	5	1.2
**Alcohol consumption**		
No	104	24.5
0.1–9.9	250	58.8
≥10	34	8.0
Missing values	37	8.7
**Menopause**		
No	122	28.7
Yes	291	68.5
Missing values	12	2.8
**Body mass index**		
<25	205	48.2
25–29.9	144	33.9
≥30	76	17.9
**Age at first menstruation**		
<12	87	20.5
12–14	274	64.5
≥15	52	12.2
**Missing values**	12	2.8

Abbreviation: SD = Standard deviation.

**Table 2 jpm-11-00858-t002:** Associations between clinicopathological factors and recurrence score category.

	Recurrence Score (RS)			
Clinicopathological Factors	Low	Intermediate	High	*p-Value*
	(N = 237)	(N = 148)	(N = 40)	
**Age**				0.1537
≤50	66 (15.5)	33 (7.8)	6 (1.4)	
>50	171 (40.2)	115 (27.1)	34 (8.0)	
**Estrogen receptor**				0.0661
1–90	23 (5.4)	18 (4.2)	9 (2.1)	
90–100	214 (50.4)	130 (30.6)	31 (7.3)	
**Progesterone receptor**				<0.0001
<1	6 (1.4)	15 (3.5)	10 (2.4)	
1–90	102 (24.0)	98 (23.1)	28 (6.6)	
90–100	129 (30.4)	35 (8.2)	2 (0.5)	
**Histological type**				0.0307
Invasive ductal carcinoma	168 (39.5)	113 (26.6)	36 (8.5)	
Other	69 (16.2)	35 (8.2)	4 (0.9)	
**Tumor stade**				0.7528
I	131 (30.8)	79 (18.6)	24 (5.6)	
II	106 (24.9)	69 (16.2)	16 (3.8)	
**Histological grade**				<0.0001
I	217 (51.1)	116 (27.3)	14 (3.3)	
II	20 (4.7)	32 (7.5)	26 (6.1)	
**Tumor size**				0.6692
≤20 mm	147 (34.6)	85 (20.0)	24 (5.6)	
>20 mm	90 (21.2)	63 (14.8)	16 (3.8)	
**Smoker status**				0.3096
Non-smoker	105 (25.0)	73 (17.4)	22 (5.2)	
Smoker	131 (31.2)	71 (16.9)	18 (4.3)	
**Alcohol consumption**				0.3768
No	60 (14.1)	35 (8.2)	9 (2.1)	
0.1–9.9	141 (33.2)	88 (20.7)	21 (4.9)	
≥10	21 (4.9)	10 (2.4)	3 (0.7)	
Missing values	15 (3.5)	15 (3.5)	7 (1.6)	
**Menopause**				0.1148
No	78 (18.9)	34 (8.2)	10 (2.4)	
Yes	154 (37.3)	109 (26.4)	28 (6.8)	
**Body mass index**				0.5218
<25	114 (26.8)	75 (17.6)	16 (3.8)	
25–29.9	76 (17.9)	52 (12.2)	16 (3.8)	
≥30	47 (11.1)	21 (4.9)	8 (1.9)	
**Age at first menstruation**				0.2344
<12	50 (12.1)	29 (7.0)	8 (1.9)	
12–14	158 (38.3)	91 (22.0)	25 (6.1)	
≥15	21 (5.1)	25 (6.1)	6 (1.5)	

**Table 3 jpm-11-00858-t003:** Multinomial multivariable logistic regressions of association between clinicopathological factors and recurrence score categories.

	Recurrence Score (RS)	
Clinicopathological Factors	Intermediate vs. Low RS	High vs. Low RS
	Adjusted OR (95% CI)	Adjusted OR (95% CI)
**Age**		
≤50	1	1
> 50	0.77 (0.33–1.82)	3.31 (0.72–15.34)
**Estrogen receptor**		
1–90	1	1
90–100	0.83 (0.41–1.71)	0.31 (0.10–0.96)
**Progesterone receptor**		
<1	3.51 (1.21–10.21)	10.34 (2.60–41.15)
1–90	1	1
90–100	0.29 (0.17–0.47)	0.05 (0.01–0.23)
**Histological type**		
Invasive ductal carcinoma	1	1
Other	0.73 (0.43–1.24)	0.36 (0.11–1.19)
**Tumor grade**		
I	1	1
II	3.16 (1.65–6.05)	23.04 (8.91–59.55)
**Menopausal**		
No	1	1
Yes	1.35 (0.59–3.13)	0.28 (0.07–1.13)

**Table 4 jpm-11-00858-t004:** Association between recurrence score and decision to have adjuvant chemotherapy.

	Recurrence Score (RS)			*p-Value*
Adjuvant Chemotherapy	Low Risk	Intermediate Risk	High Risk	<0.0001
No	234 (98.7)	85 (57.4)	3 (7.5)	
Yes	3 (1.3)	63 (42.6)	37 (92.5)	

## Data Availability

The data that support the findings of our study are available from the corresponding author upon reasonable request.

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
