# Peer review of "Identifying Clinicopathological Factors Associated with Oncotype DX® 21-Gene Recurrence Score: A Real-World Retrospective Cohort Study of Breast Cancer Patients in Quebec City, Canada"

_jpm, 2021, doi:10.3390/jpm11090858_

Round 1

Reviewer 1 Report

This study is very interesting to find a tool for use in clinical practice that helps us select patients for the determination of genomic tests. There are two ways to do it, one is the proposal in this manuscript with the use of clinical pathological parameters to find a model that can predict the result of genomic test and another is to base yourselves on your previous experience, in this sense there is a very interesting study published in JCO by Hyun-seok Kim, that has not been commented on in the article, with 1113 patients where the RS score is estimated according to the parameters based on their own results.

The main problem in this series is the low number of high-risk cases, which limits us from being able to find a prediction model for high-risk cases. Also, the way of categorizing the expression of hormone receptors is quite limited, finding only 9% negative PR. We have used the histoscore method that allows us to find a cut-off point to better categorize the PR expression.

The analysis could be done on the contrary, finding a model that predicts the cases of low vs intermediate and high RS score since they will be the ones that will avoid chemotherapy treatment and quantify the cases that are considered as low risk due to classic clinicopathological factors and afterwards, we get an intermediate-high risk score, which will be the ones who should undergo a chemotherapy treatment and we would not have done it if it were not for the genomic test.

Another very remarkable fact is the high proportion of intermediate risk cases where half of the cases receive chemotherapy treatment. It would be interesting to find a cut-off point for cases receiving chemotherapy treatment and specify the indication for such treatment in cases with a lower score.

It is also surprising that women over 50 years are more likely to intermediate and high RS and then postmenopausal women are the least likely to have high RS.

I find the time to receive the test results it is excessively long and without any justification. I think it is inadmissible for a patient to wait an average of 100 days to know if she has to undergo chemotherapy treatment.

The article provides data that helps us to predict a RS result, especially the expression of PR that is detailed in multiple studies. Despite the fact that genomic tests are implemented in all treatment guidelines, their application is still infrequent as detailed in the article (less than a third of the cases), so an effort has to be made to know how to select patients that require this test.  It is important to have predictive data from the test and to make a collaborative effort between the different centers to be capable to have more data to analyze, in this sense our center would be very enthusiastic in being able to participate and collaborate in studies of this kind.

Author Response

Response to Reviewer 1 Comments 

Comment 1. This study is very interesting to find a tool for use in clinical practice that helps us select patients for the determination of genomic tests.

OUR RESPONSE 1: We thank the reviewer for this positive and encouraging comment. 

Comment 2. There are two ways to do it, one is the proposal in this manuscript with the use of clinical pathological parameters to find a model that can predict the result of genomic test and another is to base yourselves on your previous experience, in this sense there is a very interesting study published in JCO by Hyun-seok Kim, that has not been commented on in the article, with 1113 patients where the RS score is estimated according to the parameters based on their own results.

OUR RESPONSE 2. We agree with reviewer that the work of Hyun-seok required to be added. We have added this reference to our:

  • Introduction on page 3, lines 105-106: “Histologic grade and progesterone receptor (PR) levels were the most consistent factors associated with high RS[17-21]…”
  • Discussion on page 7, line 234: “Recent American[21,31] and Asian studies[18,19] also provided findings consistent with ours.”, on page 8, lines 259 and 260: “…the variable "age" adjusted for other covariates was not significantly associated with RS, which seems to be consistent with several studies[18,20,21]…”, on page 9, lines 286 and 287: “Finally, several studies have explored the relationship between RS and Ki67 protein, some of which showed a significant relationship[18,21,38].”
  • Conclusion on page 10, lines 305-307: “Development and validation of accurate prediction models for ODX RS based on relevant clinicopathological factors are also needed in order to identify patients who should undergo a chemotherapy treatment[21].”

Comment 3. The main problem in this series is the low number of high-risk cases, which limits us from being able to find a prediction model for high-risk cases.

OUR RESPONSE: We agree with Reviewer that our sample size may have limited our study precision and capacity to characterize the high-risk cases. We indeed mention this limit in our original version of the Discussion on page 9, lines 279 and 280. In the Conclusion of this revised version (page 10, line 305), we have added the recommendation that large-scale studies are needed to consolidate our results.

Comment 4. Also, the way of categorizing the expression of hormone receptors is quite limited, finding only 9% negative PR. We have used the histoscore method that allows us to find a cut-off point to better categorize the PR expression.

OUR RESPONSE: We thank Reviewer for this information. We will investigate this method with the Deschênes-Fabia Breast Disease Center’ medical and research team.

Comment 5. The analysis could be done on the contrary, finding a model that predicts the cases of low vs intermediate and high RS score since they will be the ones that will avoid chemotherapy treatment and quantify the cases that are considered as low risk due to classic clinicopathological factors and afterwards, we get an intermediate-high risk score, which will be the ones who should undergo a chemotherapy treatment and we would not have done it if it were not for the genomic test.

OUR RESPONSE: We thank Reviewer for this suggestion. We believe however that our study does not have the parameters required to propose a predictive model per se. Our goal was to contribute to the identification of clinicopathological and lifestyle factors and we hope our findings will support future work aiming to build a decision-making tool or any kind of predictive model as it is indeed of great importance. We added the following sentence in the discussion section: “Development and validation of accurate prediction models for ODX RS based on relevant clinicopathological factors are also needed in order to identify patients who should undergo a chemotherapy treatment” (Please see page 10, lines 305-307).

Comment 6. Another very remarkable fact is the high proportion of intermediate risk cases where half of the cases receive chemotherapy treatment. It would be interesting to find a cut-off point for cases receiving chemotherapy treatment and specify the indication for such treatment in cases with a lower score.

OUR RESPONSE: We agree with this comment about the interest and usefulness of having such cut-off information to support oncologists’ medical decisions. We believe though that such assessment should be conducted in a study based on a National cancer registry coupled with an expert panel or consortium. In addition, the final decision to undergo a chemotherapy treatment lies on patients’ values and preferences. This might explain why some patients (7.5%) did not received a chemotherapy despite a high RS score as shown on table 4.

Comment 7. It is also surprising that women over 50 years are more likely to intermediate and high RS and then postmenopausal women are the least likely to have high RS.

OUR RESPONSE: We agree with Reviewer that these specific findings are surprising. We discussed these in our original version of the Discussion on page 8 and 9, lines 259 to 275. We believe this is important to report so that it can be one day pooled in a meta-analysis that would have the capacity to shed light on these particular associations.

Comment 8. I find the time to receive the test results it is excessively long and without any justification. I think it is inadmissible for a patient to wait an average of 100 days to know if she has to undergo chemotherapy treatment.

OUR RESPONSE: We thank the reviewer for raising this issue. This prompted us to have a close look on the data and a discussion with clinicians. It appears that the mean duration of 105.4 days is for the whole trajectory from the cancer diagnosis to surgery and adjuvant chemotherapy. In general, at the Deschênes-Fabia Breast Disease Center, adjuvant chemotherapy occurs within two weeks after the surgery. This accounts for the delay to receive the ODX RS. To make it clear for readers, we therefore modified the section to read as follows: “In addition, the collection of the majority of characteristics was done at baseline while the ODX test result might be received weeks later due to the process surrounding the surgery, pathology, and laboratory timeline for analysing and returning the results.” (Please see page 9, lines 280-283.

Comment 9. The article provides data that helps us to predict a RS result, especially the expression of PR that is detailed in multiple studies. Despite the fact that genomic tests are implemented in all treatment guidelines, their application is still infrequent as detailed in the article (less than a third of the cases), so an effort has to be made to know how to select patients that require this test.  It is important to have predictive data from the test and to make a collaborative effort between the different centers to be capable to have more data to analyze, in this sense our center would be very enthusiastic in being able to participate and collaborate in studies of this kind.

OUR RESPONSE: We thank reviewer for this comment and we would also be interested in collaborating with your team to pursue the next stage of this research work.

Reviewer 2 Report

The research is meaningful and satisfactorily done, but it does not reveal any new important data. The correlation between RS and clinical pathological factors such as grade and PR status is expected and described in other studies, which the authors also cite.

 Interesting and original is authors  hypothesis that lifestyle factors such as smoking, alcohol consumption and body mass index would be associated with a higher RS, which makes the manuscript innovative.
Regardless of the fact that the correlation  have not been proven, I agree with the view that it makes sense to explore further.

The authors themselves point out the limitations of the research, which would be difficult to overcome and improve for this article.

Unfortunately, the research did not offer a correlation of Rs with Ki67, which in my opinion is a shortcoming and a limitation of this research. Ki 67 is the marker that, in my opinion, could additionally correlate with the RS result.

Author Response

Response to Reviewer 2 Comments 

Comment 1. The research is meaningful and satisfactorily done…

OUR RESPONSE: We thank Reviewer for this positive and encouraging comment.

Comment 2. …, but it does not reveal any new important data. The correlation between RS and clinical pathological factors such as grade and PR status is expected and described in other studies, which the authors also cite.

OUR RESPONSE: We agree with Reviewer that our study results related to RS and clinicopathological factors are in line with what has been observed in other studies, and this is very important to report in a context where we want to accumulate evidence from different studies conducted in different locations. Contributing to the identification of factors that clearly indicate the need for chemotherapy would allow a considerable proportion of women to bypass the need for ODX testing and avoid a significant wait time before the start of their therapy. In addition, we were able for the first time to examine whether lifestyle factors are associated with ODX RS. This is clearly an original feature of our study as the reviewer himself or herself pointed out in his/her subsequent comment.

Comment 3. Interesting and original is authors hypothesis that lifestyle factors such as smoking, alcohol consumption and body mass index would be associated with a higher RS, which makes the manuscript innovative. Regardless of the fact that the correlation have not been proven, I agree with the view that it makes sense to explore further.

OUR RESPONSE: We thank the Reviewer for this comment.

Comment 4. The authors themselves point out the limitations of the research, which would be difficult to overcome and improve for this article. Unfortunately, the research did not offer a correlation of RS with Ki67, which in my opinion is a shortcoming and a limitation of this research. Ki 67 is the marker that, in my opinion, could additionally correlate with the RS result.

OUR RESPONSE: We agree that the Ki 67 would have been an important marker to include should we have had this information in our database. We do mention this point in our discussion of limitations so that readers cannot be confused about this fact.